# Genomic and Transcriptomic Characteristics According to Size of Papillary Thyroid Microcarcinoma

**DOI:** 10.3390/cancers12051345

**Published:** 2020-05-25

**Authors:** Young Shin Song, Byung-Hee Kang, Seungbok Lee, Seong-Keun Yoo, Young Sik Choi, Jungsun Park, Dong Yoon Park, Kyu Eun Lee, Jeong-Sun Seo, Young Joo Park

**Affiliations:** 1Department of Internal Medicine, CHA Bundang Medical Center, CHA University, Seongnam 13496, Korea; yssongmd@cha.ac.kr; 2Department of Biomedical Sciences, Seoul National University Graduate School, Seoul 03080, Korea; purin1985@naver.com (B.-H.K.); for3guy@naver.com (S.L.); jeongsunseo@gmail.com (J.-S.S.); 3Department of Radiation Oncology, Seoul National University College of Medicine, Seoul 03080, Korea; 4Department of Pediatrics, Seoul National University College of Medicine, Seoul 03080, Korea; 5Human Oncology and Pathogenesis Program, Memorial Sloan Kettering Cancer Center, New York, NY 10065, USA; YooS2@mskcc.org; 6Department of Internal Medicine, Kosin University College of Medicine, Busan 49267, Korea; yschoi@kosinmed.or.kr; 7Data Labs, SK Telecom, Seoul 04539, Korea; jungsun.park@sk.com (J.P.); dongyoon.park@sk.com (D.Y.P.); 8Department of Surgery, Seoul National University College of Medicine, Seoul 03080, Korea; kyueunlee@snu.ac.kr; 9Macrogen Inc., Seoul 08511, Korea; 10Department of Internal Medicine, Seoul National University College of Medicine, Seoul 03080, Korea; 11Genomic Medicine Institute, Medical Research Center, Seoul National University, Seoul 03080, Korea

**Keywords:** papillary thyroid microcarcinoma, tumor size, molecular characteristics, genome, transcriptome, massively parallel sequencing

## Abstract

It is controversial as to whether papillary thyroid microcarcinoma (PTMC) has some genomic and transcriptomic characteristics that differentiate between an early-stage lesion that would eventually evolve into the larger papillary thyroid cancer (PTC), and an occult indolent cancer in itself. To investigate this, we comprehensively elucidated the genomic and transcriptomic landscapes of PTMCs of different sizes, using a large-scaled database. This study included 3435 PTCs, 1985 of which were PTMCs. We performed targeted next-generation sequencing for 221 PTCs and integrated these data with the data including The Cancer Genome Atlas (TCGA) project. The frequency of v-raf murine sarcoma viral oncogene homolog B (*BRAF*)^V600E^ mutation was higher in PTMCs >0.5 cm than that in very small PTMCs (≤0.5 cm) and decreased again in PTCs >2 cm. Among PTMCs, the prevalence of mutations in rat sarcoma (*RAS*) and telomerase reverse transcriptase (*TERT*) promoter was not significantly different according to their size, but lower than in large PTCs. There was no change in the tumor mutational burden, the number of driver mutations, and transcriptomic profiles with tumor size, among PTMCs and all PTCs. Although a few genes with differential expression and *TERT* promoter mutations were found in a few PTMCs, our findings showed that there were no useful genomic or transcriptomic characteristics for the prediction of the future progression of PTMC.

## 1. Introduction

Papillary thyroid microcarcinoma (PTMC) is defined as a papillary thyroid cancer (PTC) measuring 1 cm or less in maximal diameter. In recent years, there has been a trend to prevent PTMC overdiagnosis and reduce the indication or extent of thyroidectomy for PTMCs, as most PTMCs show favorable prognosis [1,2]. Therefore, the 2015 American Thyroid Association guidelines have suggested active surveillance as an alternative option for PTMC without high-risk features [3]. However, some PTMCs exhibit aggressive behavior such as lateral neck lymph node and distant metastases, recurrence, and even death [4,5]. Two types of PTMCs undergo different processes, resulting in either an early-stage lesion that eventually evolves into PTC, or the formation of an occult indolent cancer in itself, and it is unclear whether they can be distinguished by their sizes alone [6]. Most of the evaluations have been based on the differences in the histological findings and the mutations between PTMC and PTC >1 cm. Unlike with histological findings, such as lymph node metastasis and extra-thyroidal extension, in most studies, no differences were observed in the proportion of major driver mutations between PTMCs and large PTCs [7,8]. However, telomerase reverse transcriptase (*TERT*) promoter mutation, known to be a strong predictor of poor prognosis in thyroid tumors, was more frequently found in large PTCs than in PTMCs [9,10], and this low frequency of *TERT* promoter mutation in PTMCs makes it difficult to compare the frequencies among PTMCs. The genomic profile was also not different among PTMCs with different statuses of lateral lymph node metastasis, whereas transcriptomic differences were found, showing 43 differentially expressed genes (DEGs) [11]. This finding suggests that molecular characteristics, especially transcriptomic changes, would be different according to the histologic characteristics associated with tumor aggressiveness. Meanwhile, the prevalence of lymph node metastasis and extra-thyroidal extension has been known to increase with an increase in PTMC tumor size [7,12], suggesting that it is important to understand the differences in the molecular characteristics of the tumor according to tumor size; however, to our knowledge, no study has investigated this yet. Therefore, in this study, we aimed to investigate the comprehensive genomic and transcriptomic landscapes, according to tumor size, even within PTMCs.

## 2. Results

### 2.1. Genomic Characteristics of PTMC Compared to PTC over 1 cm

We performed targeted next-generation sequencing for 221 PTCs (93 PTMCs and 128 PTCs >1 cm) and integrated these data with the data of non-overlapping PTC cases from three previous studies of Seoul National University Hospital (SNUH) (1853 PTMCs and 865 PTCs >1 cm) [9,13,14] and that from The Cancer Genome Atlas (TCGA) study (39 PTMCs and 457 PTCs >1 cm) [15]. In total, 3435 PTC patients (1985 PTMCs and 1450 PTCs >1 cm) were included. Clinicopathological and genomic characteristics are summarized in Table 1. PTMC had more indolent clinicopathological features than PTC >1 cm, such as a lower proportion of male patients, extrathyroidal extension, and lymph node metastasis. Moreover, in PTMC, the classical subtype was more frequent and the follicular-variant subtype was relatively rare.

V-raf murine sarcoma viral oncogene homolog B (*BRAF*)^V600E^, neuroblastoma-/Kirsten-/Harvey-rat sarcoma viral oncogene homolog (*N*/*H*/*K-RAS*), and *TERT* promoter mutations were recognized as the most frequent mutations of PTC. Compared to PTC >1 cm, PTMC had a higher prevalence of *BRAF*^V600E^ mutation (*p* < 0.001), and lower prevalence of *RAS* and *TERT* promoter mutations (*p =* 0.033 and <0.001, respectively).

Genomic characteristics other than the major mutations, including information of other mutations, tumor mutational burden (TMB), and number of driver mutations, were available only from the SNUH targeted sequencing and TCGA datasets (*n* = 221 and 496, respectively). The frequency of mutations other than *BRAF*^V600E^, *RAS*, and *TERT* promoter mutations was not different between PTMC and PTC >1 cm in both datasets. The following mutations in oncogenes or tumor suppressor genes were found in only one or two cases in the SNUH targeted sequencing dataset: KIT proto-oncogene receptor tyrosine kinase (*KIT*)^R965W^, phosphoinositide 3-kinase subunit p110-beta (*PIK3CB*)^R562Q^, epidermal growth factor receptor (*EGFR*)^L813R^, FMS-like tyrosine kinase 4 (*FLT4*)^R1060W^, ataxia telangiectasia mutated (*ATM*)^T2921M^, and neurofibromatosis type 1 (*NF1*)^W1997*^ in PTMC; *KIT*^R965W^, adenylate kinase 1 (*AKT1*)^E17K^, Janus kinase (*JAK*)*^2^*^L808W^, *ATM*^S2408L^, cyclin-dependent kinase inhibitor 2A (*CDKN2A*)^R61C^, large tumor suppressor homolog 1 (*LATS1*)^R657H^, *LATS1*^S771*^, and breast cancer gene (*BRCA*)^D932fs^ in PTC >1 cm. In TCGA dataset, rearranged during transfection (*RET*) fusions were the most frequent alterations (5.1% of PTMC and 6.8% of PTC >1 cm). The prevalence of cases in which no driver mutations were found was also similar between PTMCs and large PTCs. Moreover, TMB and number of driver mutations of PTMCs did not differ from those of PTCs >1 cm.

### 2.2. Genomic Characteristics of PTC According to Tumor Size

Next, we compared the clinicopathologic and genomic characteristics of PTCs according to tumor size after dividing by millimeter (Table 2; Figure 1). Even within PTMC, the large-sized PTMCs had high-risk clinicopathological features such as extrathyroidal extension and lymph node metastasis, despite having a high frequency of the follicular variant subtype. Among PTCs >1 cm, follicular variant subtype and lymph node metastasis were frequently observed in large tumors, whereas the frequency of extra-thyroidal extension was decreased in tumors >2 cm (Table 2). 

Because follicular variant PTC had a lower *BRAF*^V600E^ mutation rate and a higher *RAS* mutation rate than classical PTC (31.6% vs. 73.6% for *BRAF*^V600E^ and 30.0% vs. 4.1% for *RAS*), we classified PTCs by the histologic subtype (classical and follicular variant PTCs), and analyzed the prevalence of major mutations according to the tumor size in each subtype. In classical PTMC, the prevalence of *BRAF*^V600E^ mutation was significantly lower in PTMC ≤0.5 cm than that in large PTMC (*p* < 0.001; Figure 1A). The high prevalence of *BRAF*^V600E^ mutation was constant in 0.6–2 cm PTC and then decreased in PTC >2 cm (*p* for trend = 0.001). The frequency of *RAS* mutations was constant regardless of tumor size. In follicular-variant PTMC, the frequency of *BRAF*^V600E^ mutation in PTMC ≤0.5 cm was lower than that in larger PTMC, despite the lack of statistical significance, and it gradually decreased, especially in tumors >1 cm (*p* for trend <0.001; Figure 1B). The prevalence of *RAS* mutations was not significantly different according to tumor size in follicular variant PTC, as was the case in classical PTC. Among 66 of 978 tumors harboring *TERT* promoter mutations, 61 (92.4%) were PTC >1 cm and only 5 (7.6%) were PTMC (Table 1; Figure 1C,D). Interestingly, two tumors ≤0.5 cm also harbored *TERT* promoter mutations, resulting in the similar prevalence among PTMCs (3.4% in ≤0.5 cm, 1.2% in 0.6–0.7 cm, 1.6% in 0.8–1.0 cm; Table 1). However, the frequency of *TERT* promoter mutations showed a gradual increase with size in tumors >1 cm (*p* for trend <0.001; Table 2). The coexistence of *TERT* promoter and *BRAF*^V600E^ mutation was found mostly in PTCs >1 cm, except one case with the size of ≤0.5 cm. The frequency of the coexistence was increased with size among large-sized tumors of classical PTC (*p* for trend <0.001; Figure 1C). The coexistence of *TERT* promoter and *RAS* mutations showed similar findings among large-sized tumors of follicular variant PTC (Figure 1D), and there was no PTMCs in these cases.

In both the SNUH targeted sequencing and TCGA datasets, TMB did not change with tumor size, either among PTMCs or all PTCs (Figure 1E). The number of driver mutations in both datasets also did not change with the tumor size (Figure 1F). Because the alterations in mitogen-activated protein kinase (MAPK)/extracellular signal-regulated kinase (ERK) and phosphatidylinositol 3-kinase (PI3K)/protein kinase B (AKT) signaling pathways are the most frequently found and play a major role of tumorigenesis and progression in thyroid cancer [15], we analyzed the proportion of alterations in genes involved in these pathways compared to total non-silent mutations (Figure 1G,H). The average of proportions of each pathway was 32.9% and 11.4% in the SNUH targeted sequencing dataset, and 9.9% and 3.9% in TCGA dataset. When we compared this according to tumor size of PTC, there were no significant differences in both pathways.

### 2.3. Transcriptomic Characteristics of PTMC

Using the RNA sequencing database of SNUH (43 PTMCs and 81 PTCs >1 cm) and TCGA (39 PTMCs and 457 PTCs >1 cm) cohorts [14,15], we identified the transcriptomic characteristics of PTMC. Overall, principal component analysis (PCA) demonstrated that the gene expression profile was not distinguished between PTMC (various blue-colored closed circles) and PTC >1 cm (grey-colored open circles). Moreover, no distinct characteristics were found within the PTMCs according to tumor size (Figure 2A,B). Because the PTC transcriptome has been reported to be distinguishable on the basis of the mutational status of *BRAF*^V600E^ and *RAS* [10,14,15], we stratified PTCs according to these mutations. In the SNUH RNA-sequencing dataset, when we analyzed all PTCs or *RAS* mutant PTCs, there were several DEGs between PTMC and PTC >1 cm. However, these genes did not have common molecular pathways, and were difficult to consider as significant genes (Figure 2C; Table 3). Moreover, the transcriptomes of *BRAF* mutant PTCs were nearly identical between PTMC and PTC >1 cm. In TCGA dataset, the gene expression profile of PTMC was comparable with that of PTC >1 cm when we analyzed all PTCs as well as the *BRAF*-mutant or *RAS*-mutant PTCs (Figure 2D; Table 3).

Next, we classified PTMCs on the basis of their size (0.6, 0.7, and 0.8 cm) with the appropriate number of tumors in each group for analysis. We could not analyze DEGs in *RAS* mutant PTMCs due to their low prevalence (four and three PTMCs in the SNUH and TCGA datasets, respectively). Similar to the comparison between PTMC and PTC >1 cm, there were few DEGs in PTMCs of each group, which also did not have common functional pathways, in all PTMCs (Figure 3A,B; Table 3) and *BRAF* mutant PTMCs (Appendix A; Table 3) of both datasets.

*TERT* mRNA has been known as a promising prognostic marker in thyroid cancer. Nonetheless, the expression level of this gene was very low in overall PTC (baseMean, 0.66 and 2.46 in SNUH RNA-sequencing and TCGA datasets, respectively). *TERT* was not included in DEGs and its expression did not show differences according to tumor size in both datasets (Figure 3C). Furthermore, the proportion of cases with *TERT* mRNA expression also did not differ according to PTC size (Figure 3C).

To investigate whether there were differences in thyroid differentiation or MAPK activation depending on tumor size, thyroid differentiation score (TDS) and ERK score were used (Figure 3D,E). There were no changes and trends in TDS and ERK score according to tumor size, among PTMCs as well as in all PTCs, except for TDS of PTC >1 cm lower than that of PTMC in TCGA dataset alone (*p* = 0.04).

## 3. Discussion

This study showed the genomic and transcriptomic characteristics of PTC according to tumor size, particularly focusing on PTMC using an integrated large dataset of 3435 PTCs, including 1985 PTMCs from the SNUH and TCGA cohorts. PTMC was more likely to be developed by *BRAF*^V600E^ mutation, and less frequently by *RAS* or *TERT* promoter mutations, compared to PTC >1 cm. In particular, the frequency of *BRAF*^V600E^ mutation was higher in PTMC >0.5 cm than that in very small PTMC (≤0.5 cm), and decreased again in larger PTCs. The difference in the prevalence of *RAS* mutations according to tumor size disappeared when the effect of histologic subtype of PTC was excluded. That is, the frequency of *RAS* mutations was higher in large PTCs because of the higher proportion of follicular variant PTC in them. *TERT* promoter mutations, especially in coexistence with *BRAF*^V600E^ mutation, were frequently observed in larger classical PTCs, and coexistence with *RAS* mutations were only found in PTC >1 cm. There was no change in TMB and number of driver mutations with tumor size, either among PTMCs or all PTCs. Transcriptomic profiles were not distinguishable, not only between PTMC and PTC >1 cm but also among PTMCs, even when they were classified according to their size.

It is controversial as to whether PTMC could be classified into different forms: one, an early form of PTC, which will evolve into PTC on the same spectrum, and the other, an occult indolent form, which is a different subset that will not progress into PTC. To investigate this, we tried to compare the genomic and transcriptomic characteristics of PTMCs of different sizes in a large-scaled database. Overall, the genomic characteristics of the PTMCs were similar to those of 1.1–2 cm PTCs, with no differences in the prevalence of major mutations in each histologic subtype (classical and follicular variant PTCs), TMB, and the number of driver mutations, although there was a limitation in not being able to explore non-targeted variants. Regarding TMB, it has been reported to be higher in anaplastic thyroid cancer than in differentiated thyroid cancer [16], and has been associated with worse prognosis in follicular thyroid cancer [17], but there was no difference according to the various size of PTC in this study. Moreover, transcriptomic characteristics based on mRNA sequencing data also showed no significant differences among PTMCs, and even between PTMCs and larger PTCs. These results were consistent with our previous reports that PTMC behaves like a larger PTC; the prevalence of *BRAF*^V600E^ mutation and immunohistochemical staining results were not different from those obtained for PTCs [7], with no differences being observed in the gene expression profiles of PTMCs and PTCs >1 cm on the basis of microarray analysis [18]. These results suggest that the majority of the PTMCs could be representing an early state that can eventually progress to a large PTC.

In contrast, several previous studies [19,20] inferred from clinical outcomes that incidental PTMCs in particular, which were diagnosed in autopsies or after thyroidectomy secondary to benign disease, carry an excellent prognosis and do not ever evolve toward PTC. In our analysis among PTMCs, the prevalence of *BRAF*^V600E^ mutation was lower in PTMC ≤0.5 cm than that in larger PTMCs in both classical and follicular variant PTMCs, though the difference was not statistically significant in the follicular variant PTMCs. This partly supported the possibility that among very small PTMC (≤0.5 cm), an occult or indolent cancer different from the early PTC could exist. Another form, noninvasive follicular thyroid neoplasm with papillary-like nuclear features (NIFTP), a recently accepted entity, could be included as an indolent subset of PTMCs in this study. However, considering that NIFTP tends to be larger in size and had a higher frequency of *RAS* mutation at 64.7% in our study [21], the effects contaminated by NIFTPs in our PTMC subjects might be very small, although we could not completely exclude the influence. Nevertheless, to clarify these points, very small-sized tumors (≤0.5 cm) and subcentimiter non-invasive encapsulated, follicular-variant PTCs should be further investigated in future studies.

It is true that not all PTMCs may have the same fate. Recently, Perera et al. [11] presented that the genomic profiles of PTMC with or without lateral neck lymph node metastases were comparable, whereas they found 43 DEGs between them that did not overlap with DEGs identified in this study. PTMCs harboring *TERT* promoter (3.6%) and tumor protein p53 (*TP53*) (1.8%) mutations showed coexistent *BRAF*^V600E^ mutations and were restricted to the samples with lateral neck lymph node metastases. One of the four *TERT* mutants and one of the two *TP53* mutant PTMCs were 0.15 and 0.25 cm in size, respectively. In the current study, two small PTMCs (0.3 cm and 0.4 cm in size) harbored *TERT* promoter mutations, although they did not exhibit aggressive clinicopathological features such as lymph node or distant metastasis, which suggests that even PTMCs may have a potential for poor prognosis, regardless of tumor size.

Several studies have predicted molecular markers for PTMC progression [7,11,18]. In this study, although *TERT* promoter mutations in several PTMC cases and few DEGs among PTMCs were found, the frequency of *TERT* promoter mutations and the number and significance of DEGs were too low to be useful for clinical applications. Therefore, for PTMC patients undergoing active surveillance instead of thyroidectomy, there would be no useful molecular markers to predict the progression of PTMC, and regular follow-up of ultrasonography should be emphasized. However, to clarify our results, because of the retrospective design of this study, it is necessary to perform next-generation sequencing on the fine-needle biopsy samples from PTMC patients at diagnosis and the samples from those who have undergone surgery due to tumor progression in the ongoing prospective studies of active surveillance [2,22,23].

## 4. Materials and Methods

### 4.1. Patients and Tissue Samples

A total of 3435 patients with PTC (1985 PTMCs and 1450 PTCs >1 cm) were included. Formalin-fixed, paraffin-embedded tissues from 221 PTC patients were analyzed using targeted next-generation sequencing method. We integrated these data with the data obtained from four previous studies of PTC including TCGA project [9,13,14,15]. For targeted sequencing samples, cases were selected if they had residual DNA after Sanger sequencing and met the sequencing quality and quantity criteria, among samples from the previous studies of SNUH [9,13]. All tissue samples from our institution were acquired after thyroid surgery between 1993 and 2014. The pathological diagnosis was based on the seventh edition of the American Joint Committee on Cancer (AJCC) tumor-node-metastasis (TNM) staging system for differentiated thyroid cancer [24], and histological diagnosis was based on the third edition of the World Health Organization (WHO) classification for thyroid tumors [25]. The research protocol was approved by the institutional review board committee of the SNUH (no. H-1207-124-420). Information of datasets and sequencing types is summarized in Appendix A and described in detail in each reference paper.

### 4.2. Library Preparation and Sequencing

We performed targeted sequencing for 612 kinase and cancer-related genes (Appendix A). Target regions from 500 ng genomic DNA were captured using the Agilent SureSelectXT DNA Kinome Kit, following the manufacturer’s protocols (Agilent, Santa Clara, CA, USA). Briefly, DNA was sheared by the Covaris system (Covaris, Woburn, MA, USA) and purified using Agencourt AMPure XP beads (Beckman Coulter, Brea, CA, USA). The fragment ends were repaired and adaptors were ligated to the fragments. The resulting DNA library was purified using the Agencourt AMPure XP beads and amplified by polymerase chain reaction (PCR). The quality and quantity of the DNA library was assessed with the Agilent 2100 Bioanalyzer or Tapestation (D1K Tape), which was captured by hybridization to the biotinylated RNA library baits. The bound genomic DNA was purified with streptavidin-coated magnetic Dynabeads (Invitrogen, Carlsbad, CA, USA) and then re-amplified. The targeted DNA library was sequenced on Illumina Hiseq 2500 with 100 base-pair paired-end reads using recommended protocols from the manufacturer (Illumina, San Diego, CA, USA).

### 4.3. Sequence Data Analysis

The sequenced reads were aligned to the human reference genome (hg19) using the Burrows-Wheeler Aligner (BWA) [26]. The PCR duplicates were removed using Picard MarkDuplicate (http://picard.sourceforge.net) and the filtered reads were realigned at known indel positions using GATK IndelRealigner [27]. The base quality score was recalibrated by GATK BaseRecalibrator. We called single nucleotide variants (SNVs) using GATK HaplotypeCaller and MuTect [28]. GATK HaplotypeCaller was also used for indel detection. When we called variants with MuTect, tumor samples were compared against pooled normal. Variants called from these programs were used for further filters and analysis. To discover driver mutation candidates from each cancer sample, we applied additional filtration criteria as follows: (1) allele frequency <0.0001 in individuals of normal population databases of the Exome Aggregation Consortium (ExAC); (2) non-silent SNVs (nonsynonymous, splice-site) and frameshift indels; (3) predicted as a pathogenic variant, “deleterious” in variant predictor SIFT [29] and “probably damaging” or “possibly damaging” in PolyPhen-2 [30]; (4) variants annotated in COSMIC and Cancer Gene Census [31]. The variants highlighted in this study were subsequently manually reviewed. TMB was defined as the number of non-silent mutations per megabase.

### 4.4. Gene Expression Profiling, Differentially Expressed Gene Analysis, and Gene Scoring

The DEGs were determined by DESeq2 [32] to have *q*-value < 0.05, |Log_2_(fold change)| ≥ 1, and baseMean ≥ 100, and were illustrated using volcano plots, which show the magnitude and statistical significance of differential translation for each gene. The calculated *p*-values were adjusted to *q*-values for multiple testing using the Benjamini–Hochberg correction. The normalized gene expression values were applied to PCA using the most variable 500 genes. TDS and ERK scores were calculated by methods in the previous studies [14,15], and defined as the average of median-centered rlog values from DESeq2, across 16 thyroid metabolism and function genes, and 52 MAPK signaling pathway genes, respectively.

### 4.5. Statistical Analysis

Statistical analysis and visualization were performed using R statistical package. Categorical data were assessed using either the Pearson’s χ^2^ or Fisher’s exact test. Continuous data were analyzed using the independent *t*-test, Wilcoxon rank sum test, or the analysis of variance test. Binomial or multinomial logistic regression analysis was performed to access the difference in risk factors for categorical outcomes.

## 5. Conclusions

In conclusion, no specific genomic or transcriptomic characteristics were found to differ according to tumor size among PTMCs as well as all PTCs including PTC >1 cm, suggesting that a large portion of PTMCs exist in an early form, which would eventually progress to form a large PTC. Although a few genes with differential expression and *TERT* promoter mutations were found in a few PTMCs, no useful genomic or transcriptomic characteristics were found for the prediction of the future progression of PTMC, bearing important implications for PTMC patients on active surveillance. 

## Figures and Tables

**Figure 1 cancers-12-01345-f001:**
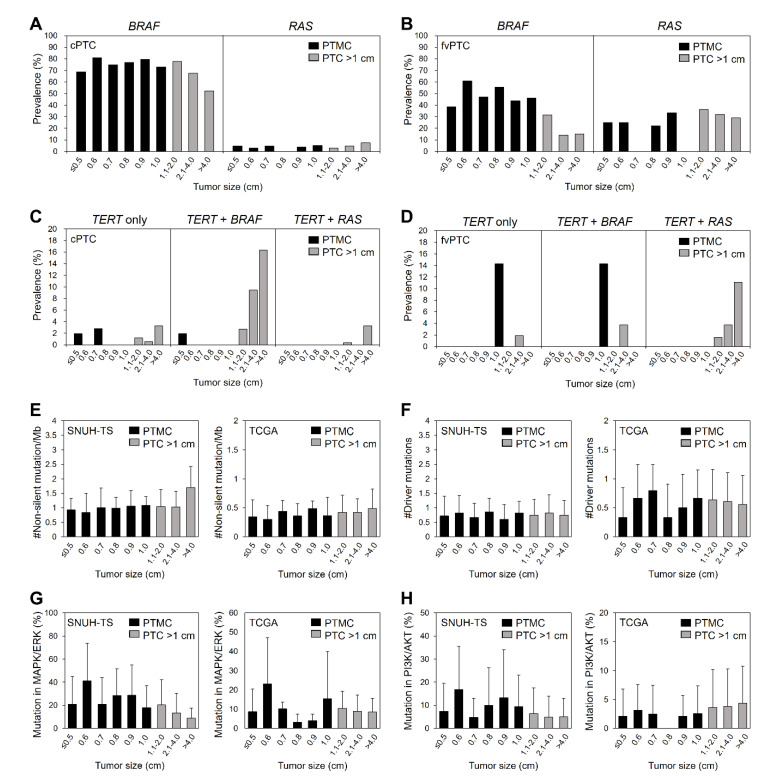
Genomic characteristics of papillary thyroid carcinoma according to tumor size. Prevalence of *BRAF*^V600E^ and *RAS* mutations according to tumor size in (**A**) classical (*n* = 2954 and 851 for *BRAF*^V600E^ and *RAS*, respectively) and (**B**) follicular variant (*n* = 342 and 217 for *BRAF*^V600E^ and *RAS*, respectively) PTCs. Prevalence of *TERT* promoter mutations (*TERT* alone, *TERT* + *BRAF*, and *TERT* + *RAS*) according to tumor size in (**C**) classical (*n* = 707) and (**D**) follicular variant (*n* = 184) PTCs. (**E**) Tumor mutational burden, (**F**) number of driver mutations, and (**G**) proportions of non-silent mutations involved in mitogen-activated protein kinase (MAPK)/extracellular signal-regulated kinase (ERK) pathway and (**H**) in phosphatidylinositol 3-kinase (PI3K)/protein kinase B (AKT) pathways compared to total non-silent mutations, according to tumor size of PTC in the SNUH targeted sequencing (SNUH-TS, *n* = 221) and TCGA (*n* = 496) datasets. cPTC, classical papillary thyroid carcinoma; fvPTC, follicular-variant papillary thyroid carcinoma; PTMC, papillary thyroid microcarcinoma; PTC, papillary thyroid carcinoma.

**Figure 2 cancers-12-01345-f002:**
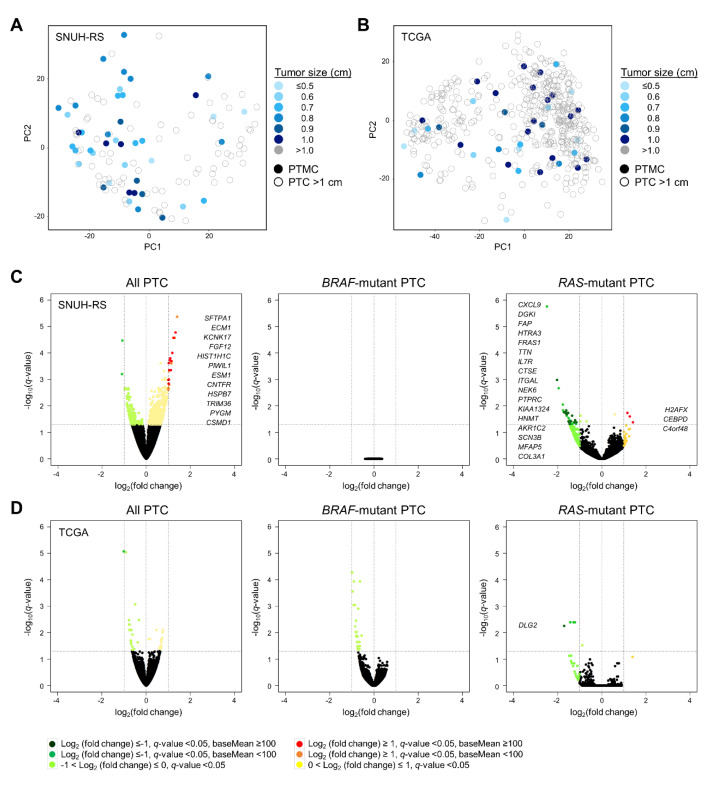
Transcriptomic characteristics of papillary thyroid microcarcinoma. Principal component analysis plot of PTC from the (**A**) SNUH RNA-sequencing (SNUH-RS, *n* = 124) and (**B**) TCGA (*n* = 496) datasets. Volcano plots showing differential RNA expression levels between PTMC and PTC >1 cm in all PTCs (left), *BRAF* mutant PTCs (middle), and *RAS* mutant PTCs (right) from the (**C**) SNUH-RS (*n* = 43 vs. 81, 29 vs. 37, 4 vs. 20, respectively) and (**D**) TCGA (*n* = 39 vs. 457, 18 vs. 217, 3 vs. 49, respectively) datasets. Up- and down-regulated DEGs in PTC >1 cm compared to PTMC are marked in dark red and dark green, respectively. PTMC, papillary thyroid microcarcinoma; PTC, papillary thyroid carcinoma.

**Figure 3 cancers-12-01345-f003:**
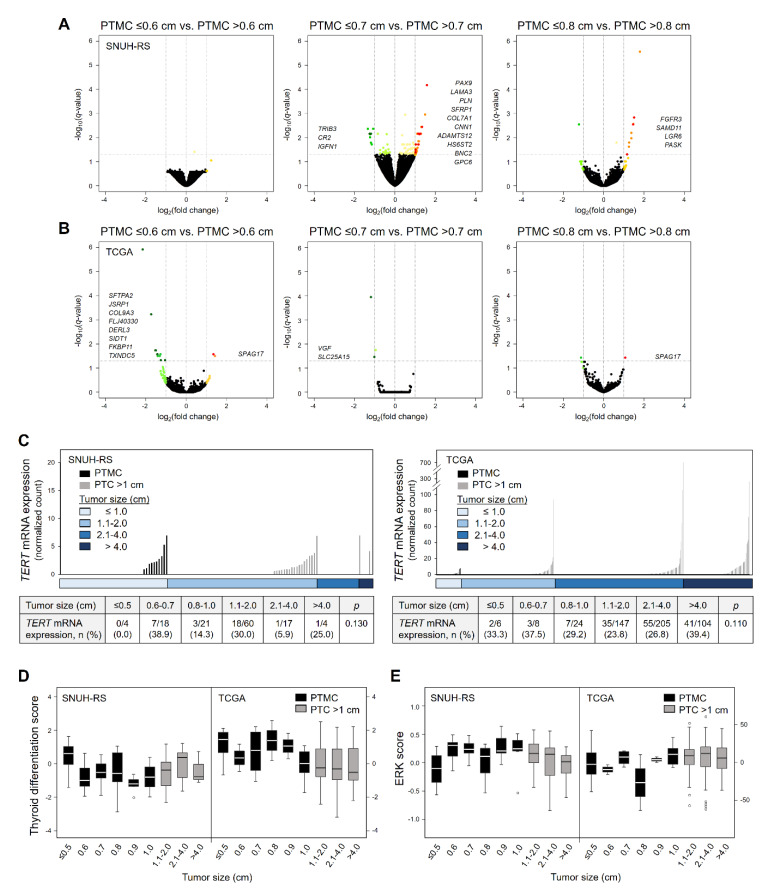
Transcriptomic characteristics of papillary thyroid microcarcinoma according to tumor size. Volcano plots showing differential RNA expression levels between groups of all PTMCs: ≤0.6 cm vs. >0.6 cm (left), ≤0.7 cm vs. >0.7 cm (middle), and ≤0.8 cm vs. >0.8 cm (right) from the (**A**) SNUH RNA-sequencing (SNUH-RS, *n* = 13 vs. 30, 22 vs. 21, 32 vs. 11, respectively) and (**B**) TCGA (*n* = 9 vs. 30, 14 vs. 25, 17 vs. 22, respectively) datasets. (**C**) *TERT* mRNA expression levels according to tumor size (upper graph) from both datasets. Each column represents an individual sample. The number of cases with *TERT* mRNA expression according to tumor size (lower table). Thyroid differentiation score (**D**) and ERK score (**E**) according to tumor size from both datasets (*n* = 124 and 496 for SNUH-RS and TCGA datasets, respectively). PTMC, papillary thyroid microcarcinoma.

**Table 1 cancers-12-01345-t001:** Clinicopathological and genomic characteristics of papillary thyroid microcarcinoma (PTMC) and papillary thyroid carcinoma (PTC) >1 cm.

Characteristics	PTMC	PTC > 1 cm	*p*
No. of patients	1985	1450	
Age, mean ± SD	47.6 ± 11.5	47.6 ± 14.6	0.994
Male sex, *n* (%)	332/1985 (16.7)	314/1450 (21.7)	<0.001
Tumor size, cm, median	0.6 (0.5–0.8)	1.7 (1.3–2.5)	<0.001
PTC subtype, *n* (%)			<0.001
Classical	1828/1964 (93.1)	1126/1417 (79.5)	
Follicular-variant	113/1964 (5.8)	229/1417 (16.2)	
Extrathyroidal extension, *n* (%)	1003/1964 (51.1)	882/1445 (61.0)	<0.001
Minimal	951/1964 (48.4)	656/1445 (45.4)	<0.001
Gross	52/1964 (2.6)	226/1445 (15.6)	
LN metastasis, *n* (%)	514/1980 (26.0)	690/1391 (49.6)	<0.001
Major mutation, *n* (%)			
*BRAF*^V600E^	1438/1985 (72.4)	941/1450 (64.9)	<0.001
*RAS*	16/309 (5.2)	84/844 (10.0)	0.011
*TERT*	5/266 (1.9)	61/712 (8.6)	<0.001
Other drivers, *n* (%) ^1^	6/93 (6.5) ^2^; 14/39 (35.9) ^3^	8/128 (6.3) ^2^; 127/457 (27.8) ^3^	0.952 ^2^; 0.281 ^3^
No driver, *n* (%)	28/93 (30.1) ^2^; 4/39 (10.3) ^3^	39/128 (30.5) ^2^; 60/457 (13.1) ^3^	0.954 ^2^; 0.607 ^3^
TMB, mean ± SD	0.97 ± 0.48 ^2^; 0.38 ± 0.26 ^3^	1.06 ± 0.59 ^2^; 0.44 ± 0.28 ^3^	0.262 ^2^; 0.272 ^3^
No. of driver mutations, mean ± SD	0.75 ± 0.54 ^2^; 0.59 ± 0.50 ^3^	0.77 ± 0.57 ^2^; 0.61 ± 0.51 ^3^	0.865 ^2^; 0.846 ^3^

^1^ Other driver mutations rather than major mutations such as v-raf murine sarcoma viral oncogene homolog B (*BRAF*)^V600E^, rat sarcoma (*RAS*), and telomerase reverse transcriptase (*TERT*) promoter mutations; ^2^ values from Seoul National University Hospital (SNUH) targeted sequencing dataset; ^3^ values from The Cancer Genome Atlas (TCGA) dataset. For categorical variables, the number of denominators of each variable varied on the basis of the number of subjects who had the information. SD, standard deviation; LN, lymph node; TMB, tumor mutational burden defined as non-silent mutations per megabase.

**Table 2 cancers-12-01345-t002:** Clinicopathological and genomic characteristics of papillary thyroid carcinoma (PTC) according to tumor size.

Characteristics	≤0.5 cm	0.6–0.7 cm	0.8–1.0 cm	1.1–2.0 cm	2.1–4.0 cm	>4.0 cm	*p* ^1^	*p* ^2^
No. of patients	752	580	653	906	410	134		
Age, mean ± SD	47.3 ± 11.4	47.8 ± 11.6	47.7 ± 11.6	48.3 ± 13.5	44.9 ± 15.8	50.4 ± 17.0	0.401	0.983
Male sex, *n* (%)	117 (15.6)	94 (16.2)	121 (18.5)	172 (19.0)	93 (22.7)	49 (36.6)	0.282	<0.001
PTC subtype, *n* (%)							<0.001	<0.001
Classical	712/748 (95.6)	535/577 (92.7)	581/642 (90.5)	737/887 (83.1)	301/398 (75.6)	88/132 (66.7)		
Follicular variant	26/748 (3.5)	40/577 (6.9)	47/642 (7.3)	118/887 (13.3)	78/398 (19.6)	33/132 (25.0)		
Extrathyroidal extension, *n* (%)	247/735 (33.6)	319/578 (55.2)	437/651 (67.1)	612/903 (67.8)	207/409 (50.6)	63/133 (47.4)	<0.001	<0.001
Minimal	239/735 (32.5)	306/578 (52.9)	406/651 (62.4)	464/903 (51.4)	147/409 (35.9)	45/133 (33.8)	<0.001	<0.001
Gross	8/735 (1.1)	13/578 (2.2)	31/651 (4.8)	148/903 (16.4)	60/409 (14.7)	18/133 (13.5)		
LN metastasis, *n* (%)	126/750 (16.8)	175/578 (30.3)	213/652 (32.7)	424/890 (47.6)	198/378 (52.4)	68/123 (55.3)	<0.001	<0.001
Major mutation, *n* (%)								
*BRAF*^V600E^	507/752 (67.4)	443/580 (76.4)	488/653 (74.7)	647/906 (71.4)	235/410 (57.3)	59/134 (44.0)	0.032	<0.001
*RAS*	4/71 (5.6)	5/95 (5.3)	7/143 (4.9)	37/406 (9.1)	32/315 (10.2)	15/123 (12.2)	0.827	0.011
*TERT*	2/59 (3.4)	1/84 (1.2)	2/123 (1.6)	14/353 (4.0)	29/260 (11.2)	18/99 (18.2)	1.000	<0.001

^1^*p*-values for trend according to tumor size in papillary thyroid microcarcinoma (≤1.0 cm); ^2^*p*-values for trend according to tumor size in all papillary thyroid carcinoma. SD, standard deviation; LN, lymph node.

**Table 3 cancers-12-01345-t003:** Differentially expressed genes between papillary thyroid microcarcinoma (PTMC) and papillary thyroid carcinoma (PTC) >1 cm, and according to PTMC size.

Subject	SNUH-RS Dataset ^1^			TCGA Dataset ^2^		
	Gene	*q*-Value	Log_2_(FC)	Gene	*q*-Value	Log_2_(FC)
All PTC						
≤1.0 vs. >1.0 cm	*SFTPA1*	1.67 × 10^−5^	1.33			
(No. ^1^ = 43 vs. 81 No. ^2^ = 39 vs. 457)	*ECM1*	2.67 × 10^−5^	1.25			
	*KCNK17*	9.92 × 10^−5^	1.18			
	*FGF12*	1.58 × 10^−4^	1.09			
	*HIST1H1C*	1.84 × 10^−4^	1.06			
	*PIWIL1*	2.02 × 10^−4^	1.13			
	*ESM1*	2.40 × 10^−4^	1.02			
	*CNTFR*	4.45 × 10^−4^	1.12			
	*HSPB7*	4.45 × 10^−4^	1.05			
	*TRIM36*	1.01 × 10^−3^	1.03			
	*PYGM*	1.03 × 10^−3^	1.00			
	*CSMD1*	1.45 × 10^−3^	1.02			
*RAS* mutant PTC						
≤1.0 vs. >1.0 cm	*CXCL9*	1.03 × 10^−3^	−2.02	*DLG2*	5.55 × 10^−3^	−1.70
(No. ^1^ = 4 vs. 20 No. ^2^ = 3 vs. 49)	*DGKI*	1.46 × 10^−2^	−1.61			
	*FAP*	1.75 × 10^−2^	−1.71			
	*HTRA3*	1.75 × 10^−2^	−1.68			
	*FRAS1*	1.83 × 10^−2^	−1.60			
	*TTN*	1.83 × 10^−2^	−1.52			
	*IL7R*	1.83 × 10^−2^	−1.62			
	*CTSE*	2.08 × 10^−2^	−1.59			
	*ITGAL*	2.33 × 10^−2^	−1.34			
	*NEK6*	3.44 × 10^−2^	−1.21			
	*PTPRC*	3.77 × 10^−2^	−1.39			
	*KIAA1324*	3.84 × 10^−2^	−1.51			
	*HNMT*	4.15 × 10^−2^	−1.15			
	*AKR1C2*	4.15 × 10^−2^	−1.48			
	*SCN3B*	4.15 × 10^−2^	−1.46			
	*MFAP5*	4.15 × 10^−2^	−1.23			
	*COL3A1*	4.77 × 10^−2^	−1.45			
	*H2AFX*	1.83 × 10^−2^	1.16			
	*CEBPD*	2.50 × 10^−2^	1.27			
	*C4orf48*	4.15 × 10^−2^	1.41			
All PTMC						
≤0.6 vs. >0.6 cm				*SFTPA2*	1.21 × 10^−6^	−2.16
(No. ^1^ = 13 vs. 30 No. ^2^ = 9 vs. 30)				*JSRP1*	5.94 × 10^−4^	−1.74
				*COL9A3*	1.85 × 10^−2^	−1.50
				*FLJ40330*	1.85 × 10^−2^	−1.53
				*DERL3*	2.68 × 10^−2^	−1.44
				*SIDT1*	2.68 × 10^−2^	−1.43
				*FKBP11*	4.72 × 10^−2^	−1.26
				*TXNDC5*	4.72 × 10^−2^	−1.05
				*SPAG17*	2.67 × 10^−2^	1.35
≤0.7 vs. >0.7 cm	*TRIB3*	6.96 × 10^−6^	−1.19	*VGF*	1.14 × 10^−4^	−1.19
(No. ^1^ = 22 vs. 21 No. ^2^ = 14 vs. 25)	*CR2*	6.96 × 10^−6^	−1.26	*SLC25A15*	3.45 × 10^−2^	−1.02
	*IGFN1*	7.21 × 10^−6^	−1.23			
	*PAX9*	6.67 × 10^−5^	1.58			
	*LAMA3*	3.59 × 10^−3^	1.34			
	*PLN*	6.88 × 10^−3^	1.15			
	*SFRP1*	6.96 × 10^−3^	1.26			
	*COL7A1*	6.96 × 10^−3^	1.24			
	*CNN1*	7.21 × 10^−3^	1.21			
	*ADAMTS12*	1.92 × 10^−2^	1.03			
	*HS6ST2*	1.92 × 10^−2^	1.16			
	*BNC2*	3.23 × 10^−2^	1.02			
	*GPC6*	4.07 × 10^−2^	1.02			
≤0.8 vs. >0.8 cm	*FGFR3*	2.45 × 10^−7^	1.87	*SPAG17*	3.69 × 10^−2^	1.07
(No. ^1^ = 32 vs. 11 No. ^2^ = 17 vs. 22)	*SAMD11*	1.47 × 10^−3^	1.51			
	*LGR6*	2.83 × 10^−3^	1.46			
	*PASK*	4.95 × 10^−2^	1.16			
*BRAF* mutant PTMC						
≤0.6 vs. >0.6 cm				*TMEM132D*	1.04 × 10^−5^	−1.81
(No. ^1^ = 11 vs. 18 No. ^2^ = 3 vs. 15)				*FOXA2*	4.54 × 10^−3^	−1.06
				*DDO*	3.62 × 10^−2^	1.18
≤0.7 vs. >0.7 cm	*LAMA3*	3.44 × 10^−3^	1.17	*KIAA1199*	6.96 × 10^−3^	−1.07
(No. ^1^ = 18 vs. 11 No. ^2^ = 6 vs. 12)	*PHEX*	3.44 × 10^−3^	1.22			
	*GLDN*	4.65 × 10^−3^	1.08			
	*HS6ST2*	1.79 × 10^−2^	1.04			
	*PLN*	2.83 × 10^−2^	1.02			
≤0.8 vs. >0.8 cm				*MKX*	1.52 × 10^−2^	−1.24
(No. ^1^ = 22 vs. 7 No. ^2^ = 7 vs. 11)				*KIAA1199*	1.58 × 10^−2^	−1.12


^1^ SNUH-RNA sequencing (SNUH-RS) dataset from Yoo, S.K.; Lee, S.; Kim, S.J.; Jee, H.G.; Kim, B.A.; Cho, H.; Song, Y.S.; Cho, S.W.; Won, J.K.; Shin, J.Y., et al. Comprehensive analysis of the transcriptional and mutational landscape of follicular and papillary thyroid cancers. *PLoS Genet*
**2016**, *12*, e1006239; ^2^ TCGA dataset from Cancer Genome Atlas Research Network. Integrated genomic characterization of papillary thyroid carcinoma. *Cell*
**2014**, *159*, 676–690. FC, fold change.

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
