# Peer review of "Genomic and Transcriptomic Characteristics According to Size of Papillary Thyroid Microcarcinoma"

_cancers, 2020, doi:10.3390/cancers12051345_

Round 1
Reviewer 1 Report
Dr. Song and colleagues have performed targeted NGS analyses of a significant number of PTCs of various sizes, as well as integrated these findings with present data from established databases (including NGS and transcriptome data). The aim of the study was to identify key genetic alterations in papillary thyroid microcarcinomas (PTMCs) and try to couple these findings to clinical parameters, to single out cases exhibiting poor prognosis, and to compare these findings with the results from larger (>10 mm) cases. The authors conclude that neither BRAF V600E or TERT promoter mutations can be used to foresee poor outcome cases among these sub-centimeter cases, nor can mutational burden or expressional profiles. Although partly confirmatory in nature, the sheer number of tumors gives this study power. I have several suggestions for the authors to consider:
1: In Table 1, more than 50% of cases in the PTMC cohort exhibit extrathyroidal extension. This is remarkable, as PTMCs rarely grows into the strap muscle. In other studies, the rate of ETE in PTMC is much lower, between 5-10% (see for example PMID: 20052467). Please comment on this, and also explain how ETE was defined, as I assume the cohort spans over two different WHO (2004 and 2017) and TNM (v7 and v8) classification periods?
2: Speaking of the WHO classifications, have the authors considered stratifying their material into the newly accepted entity NIFTP? It might not be possible to perform practically given the large number of cases, but would it be wrong to assume that many of the PTMCs included in this study might actually fit the criteria for an NIFTP diagnosis? I urge the authors to discuss this, as it might influence the outcome of the study (subsets of PTMCs with a follicular growth pattern and lack of invasive attributes et.c. might be more indolent than corresponding tumors that actually invade the tumor capsule).
3: Tumor mutational burden is detailed in Table 1, but the values are not defined. Mutations/Mb? Please make this clear. Also, mutational burden has been significantly associated to worse prognosis in follicular thyroid cancer, this could be relevant for the authors to comment on as they seek this association for PTCs. (PMID 29726952).
4: Overall, the introduction section is too short and superficial. The authors should mention that TERT promoter mutations are strong predictors of recurrence in both "benign" (FT-UMPs, follicular adenomas) and malignant thyroid lesions, and amend the appropriate references. Thus, the reader will grasp the importance of this marker to a greater extent.
5. The role of TERT mRNA in predicting worse outcome in thyroid cancer is gaining ground, but the authors do not mention this gene. As it was not included in Table 3 (including significant DEGs), it would be interesting to see what the authors make of this - especially since TERT mRNA has been found prognostic in PTMCs, PTCs as well as in follicular thyroid carcinomas (PMID: 31286848 and PMID: 29692346 to name a few). The discussion could be expanded as of this.
6: TERT promoter mutations are designated as "driver mutations" by the authors (page 2, line 81). I am not sure this is well established, the mutations are more likely coupled to progression rather than oncogenic driver events. Please revise. Also, speaking of progression - the authors' model strongly argues for additional genetic events governing the poor outcome in certain PTC cases besides from the classical genetic hits, as small and larger PTCs were largely comparable. Additional events in non-coding DNA could therefore be expected, as I assume the authors employ WES data and not WGS data? Please revise the discussion appropriately.
Reviewer 2 Report
Song et al in their manuscript titled “Genomic and Transcriptomic Characteristics According to Size of Papillary Thyroid Microcarcinoma” elucidated the genomic and transcriptomic landscapes of PTMCs of different sizes, using a large-scaled database including 3435 PTCs, 1985 of which were PTMCs. They performed targeted next-generation sequencing for 221 PTCs and integrated this data with the data from TCGA project. They found that the frequency of BRAFV600E mutation was higher in PTMCs >0.5 cm than that in very small PTMCs (≤0.5 cm) and decreased again in PTCs >2 cm. Among PTMCs, the prevalence of mutations in RAS and TERT promoter was not significantly different according to their size, but lower than in large PTCs. There was no change in the tumor mutational burden, the number of driver mutations, and transcriptomic profiles with tumor size, among PTMCs and all PTCs.
They conclude that though a few genes with differential expression and TERT promoter mutations were found in a few PTMCs, their findings showed that there were no useful genomic or transcriptomic characteristics for the prediction of the future progression of PTMC.
The authors did an interesting work with relevant experiments. I have the following comments.
What does the authors mean here
They found that the frequency of BRAFV600E mutation was higher in PTMCs >0.5 cm than that in very small PTMCs (≤0.5 cm) and decreased again in PTCs >2 cm. is it <2 cm or >2 cm? If it is >2cm then it should be increased
From line 59 to 63 the authors wrote that “Unlike with histological findings, such as lymph node metastasis and extra-thyroidal extension, no differences were observed in the proportion of major driver mutations between PTMCs and large PTCs, in most studies [7,8]. A recent study also showed
no significant genomic differences between PTMCs with and without lateral lymph node metastases, but transcriptomic differences with 43 differentially-expressed genes (DEGs) [9]. What does this mean? There were no mutations but only transcription level differences? Did the authors check for mutations and gene expression changes in these genes?
From line 123 to 127 the authors write that “Most tumors harboring TERT promoter mutations were PTC >1 cm, although three cases of classical PTMC and two cases of follicular-variant PTMC had these mutations (Figures 1C and 1D). Interestingly, two patients with classical PTC ≤0.5 cm had TERT promoter mutations. In particular, the coexistence of TERT promoter and BRAFV600E mutations were frequently found in large-sized tumors of classical PTC (p for trend 1 cm (Figure 1D).” Does this mean the effect of TERT mutations is only in combination with BRAF mutations?
From fig 1E to F there is no difference in tumor size and number of driver mutations between PTMC and PTC. What makes PTC aggressive compared to PTMC then?
From line 177 to 181 the authors wrote that “To investigate whether there were differences in thyroid differentiation or mitogen-activated protein kinase (MAPK) activation depending on tumor size, thyroid differentiation score (TDS) and extracellular signal-regulated kinase (ERK) score were used (Figures 3C and 3D). There were no changes and trends in TDS and ERK score according to tumor size, among PTMCs as well as in all PTCs, except for lower TDS of PTC >1 cm than that of PTMC in only TCGA dataset (p = 0.04)”. In this context the authors should refer to a recent paper in biomolecules “Whole Genome Sequencing of Familial Non-Medullary Thyroid Cancer Identifies Germline Alterations in MAPK/ERK and PI3K/AKT Signaling Pathways.” Where they have showed the mutations in genes involved in mitogen-activated protein kinase (MAPK) and extracellular signal-regulated kinase (ERK) and see if there is any correlation.
Round 2
Reviewer 1 Report
The authors have responded excellent to my previous concerns.